# RL-LIM: Reinforcement Learning-based Locally Interpretable Modeling

## Abstract

Understanding black-box machine learning models is important towards their widespread adoption. However, developing globally interpretable models that explain the behavior of the entire model is challenging. An alternative approach is to explain black-box models through explaining individual prediction using a locally interpretable model. In this paper, we propose a novel method for locally interpretable modeling – Reinforcement Learning-based Locally Interpretable Modeling (RL-LIM). RL-LIM employs reinforcement learning to select a small number of samples and distill the black-box model prediction into a low-capacity locally interpretable model. Training is guided with a reward that is obtained directly by measuring agreement of the predictions from the locally interpretable model with the black-box model. RL-LIM near-matches the overall prediction performance of black-box models while yielding human-like interpretability, and significantly outperforms state of the art locally interpretable models in terms of overall prediction performance and fidelity.

## 1 Introduction

Artificial Intelligence (AI) is advancing at a rapid pace, particularly with recent advances in deep neural networks and ensemble methods (Goodfellow et al., 2016; He et al., 2016; Chen & Guestrin, 2016; Ke et al., 2017). This progress has been fueled by 'black-box' machine learning models where the decision making is controlled by complex non-linear interactions between many parameters that are difficult for humans to understand and interpret. However, in many real-world applications AI systems are not only expected to perform well but are also required to be interpretable: doctors need to understand why a particular treatment is recommended, and financial institutions need to understand why a loan was declined. Use cases of model interpretability vary across applications: it can provide trust to users by showing rationales behind decisions, enable detection of systematic failure cases, and provide actionable feedback for improving models (Rudin, 2018).

Many studies have suggested a trade-off between performance and interpretability (Virág & Nyitrai, 2014; Johansson et al., 2011). This is correct in that globally interpretable models, which attempt to explain the entire model behavior, typically yield considerably worse performance than 'black-box' models (Lipton, 2016). To go beyond the performance limitations of globally interpretable models, another promising direction is locally interpretable models, which instead of explaining the entire model explain a single prediction (Ribeiro et al., 2016). Methodologically, while a globally interpretable model fits a single inherently interpretable model (such as a linear model or a shallow decision tree) to the entire training set, locally interpretable models aim to fit an inherently interpretable model locally, i.e. for each instance individually, by distilling knowledge from a high performance black-box model. Such locally interpretable models are very useful for real-world AI deployments to provide succinct and human-like explanations to users. They can be used to identify systematic failure cases (e.g. by seeking common trends in input dependence for failure cases), detect biases (e.g. by quantifying feature importance for a particular variable), and provide actionable feedback to improve a model (e.g. understand failure cases and what training data to collect).

To be useful in practice, locally interpretable models need to maximize two objectives: (i) the overall prediction performance (how well it predicts compared to the ground truth labels) – for the model to be accurate, and (ii) fidelity (how well it approximates the 'black-box' model predictions) – to ensure the model is reliably approximating the black-box model's predictions in the neighborhood

of interest (Plumb et al., 2019; Lakkaraju et al., 2019). To this end, a few methods have recently been proposed for locally interpretable modeling: Local Interpretable Model-agnostic Explanations (LIME) (Ribeiro et al., 2016), Supervised Local modeling methods (SILO) (Bloniarz et al., 2016), and Model Agnostic Supervised Local Explanations (MAPLE) (Plumb et al., 2018). LIME in particular has gained notable popularity and has been deployed in many applications due to its simplicity. However, the overall prediction performance and fidelity metrics are not reaching desired levels in many cases (Alvarez-Melis & Jaakkola, 2018; Zhang et al., 2019; Ribeiro et al., 2018; Lakkaraju et al., 2017). Indeed, as we show in our experiments, there are frequent cases where existing locally interpretable models even underperform commonly low-performing globally interpretable models.

One of the fundamental challenges to fit a locally interpretable model is the representational capacity difference while applying distillation. Black-box machine learning models, such as deep neural networks or ensemble models, have much larger representational capacity than locally interpretable models. This can result in underfitting with conventional distillation techniques, leading to suboptimal performance (Hinton et al., 2015; Wang et al., 2019). We address this fundamental challenge by proposing a novel Reinforcement Learning-based method to fit Locally Interpretable Models which we call RL-LIM. RL-LIM efficiently utilizes the small representational capacity of locally interpretable models by training with a small number of samples that are determined to have the highest value contribution to the fitting of a locally interpretable model. In order to select these highest-value instances, we train instance-wise weight estimators (modeled with deep neural networks) using a reinforcement signal that quantifies the fidelity metric (i.e. how well does the model approximate the black-box model predictions). The contributions of this paper can be summarized as:

1. We introduce the first method that tackles interpretability through data-weighted training, and show that reinforcement learning is highly effective for end-to-end training of such a model.
2. We show that distillation of a black-box model into a low-capacity interpretable model can be significantly improved by fitting with a small subset of relevant samples that is controlled efficiently by our method.
3. On various classification and regression datasets, we demonstrate that RL-LIM significantly outperforms alternative models (LIME, SILO and MAPLE) in overall prediction performance and fidelity metrics – in most cases, the overall performance of locally interpretable models obtained by RL-LIM is very similar to complex black-box models.

## 2 RELATED WORK

**Locally interpretable models:** There are various approaches to interpret black-box models – (Gilpin et al., 2018) provides a good overview. One approach is to directly decompose the prediction into feature attributions by considering what-if cases. Shapley values (Štrumbelj & Kononenko, 2014) and their computationally-efficient variants (Lundberg & Lee, 2017) are commonly-used methods in this category. Other notable methods are based on activation differences, e.g. DeepLIFT (Shrikumar et al., 2017), or saliency maps using the gradient flows, e.g. CAM (Zhou et al., 2016) and Grad-CAM (Selvaraju et al., 2017). In this paper, we focus on the direction of locally interpretable modeling – distilling a black-box model into an interpretable model for each input instance.

Locally Interpretable Model-agnostic Explanation (LIME) (Ribeiro et al., 2016) is the most popular method for locally interpretable modeling. LIME is based on modifying a data instance by tweaking the feature values and then learning from the impact of the modifications on the output. A fundamental challenge for LIME is the need for a meaningful distance metric to determine neighborhoods, as simple metrics like Euclidean distance may yield poor fidelity in some cases and the estimation can be highly-sensitive to normalization (Alvarez-Melis & Jaakkola, 2018) especially with categorical variables. Supervised Local modeling methods (SILO) (Bloniarz et al., 2016)) aims to improve LIME by determining the neighborhoods for each instance using ad-hoc tree-based ensemble methods. Model Agnostic Supervised Local Explanations (MAPLE) (Plumb et al., 2018) furthers adds a method for feature selection on top of SILO – it utilizes ad-hoc tree-based ensemble methods to determine the weights of training instances for each target instance and uses the weights to optimize a locally interpretable model. However, SILO and MAPLE still have shortcomings because the tree-based ensemble methods are optimized independently from the locally interpretable model – lack of joint optimization results in suboptimal fidelity for the locally interpretable model. Overall, to construct a locally interpretable model, a key problem is how to select the optimal training

instances for each testing instance, because the selected training instances mostly determine the constructed locally interpretable model. The number of possibilities for training instance selection is extremely large (exponential in the number of training instances). LIME heuristically utilizes Euclidean distances, whereas SILO and MAPLE use ad-hoc tree-based ensemble methods. Our proposed method, RL-LIM, takes a very different perspective: to properly and efficiently explore the large possible solution space, RL-LIM utilizes reinforcement learning to find the optimal policy that selects the training instances that maximize the fidelity of the locally interpretable model.

**Data-weighted training:** Optimal weighing of training data is a paramount problem in machine learning. By upweighting valuable instances and downweighting the low quality or problematic instances, better performance can be obtained in certain learning scenarios, such as imbalanced or noisy labels (Jiang et al., 2018). One approach for data weighting is utilizing Influence Functions (Koh & Liang, 2017), that are based on oracle access to gradients and Hessian-vector products. Jointly-trained student-teacher methods constitute another approach (Jiang et al., 2018; Bengio et al., 2009) to learn a data-driven curriculum. Using the feedback from the teacher network, training instance-wise weights are learned for the student model. Aligned with our motivations, meta learning is considered for data weighting in Ren et al. (2018). Their proposed method utilizes gradient descent-based meta learning, guided by a small validation set, to maximize the target performance.

In this work we consider data-weighted training for a novel purpose: interpretability. Unlike gradient descent-based meta learning, our approach uses reinforcement learning to integrate the reward directly with the fidelity metric. Aforementioned works estimate the same ranking of training instances for the entire dataset. Instead, our method yields an instance-wise ranking of training data points, different for each testing instance. This enables efficient distillation of a black-box model prediction into a locally interpretable model.

## 3 REINFORCEMENT LEARNING-BASED MODELING

We consider a training dataset $\mathcal{D} = \{(\mathbf{x}_i, y_i), i = 1, ..., N\} \sim \mathcal{P}$ for training of a black-box model $f$, where $\mathbf{x}_i \in \mathcal{X}$ is the feature vector in a $d$-dimensional feature space $\mathcal{X}$ and $y_i \in \mathcal{Y}$ is the corresponding label in a label space $\mathcal{Y}$. We also assume that there exists a probe dataset $\mathcal{D}^p = \{(\mathbf{x}_j^p, y_j^p), j = 1, ..., M\} \sim \mathcal{P}$ where $M$ is the number of probe instances. The probe dataset is used to evaluate the model performance to guide meta-learning as in Ren et al. (2018). If there is no explicit probe dataset, we can randomly partition a subset of the training dataset as the probe dataset and the remainder as the training dataset. RL-LIM is composed of three models:

1. **Black-box model** $f : \mathcal{X} \to \mathcal{Y}$ – any machine learning model that needs to be explained (e.g. a deep neural network or a decision tree-based ensemble model),

2. **Locally interpretable model** $g_\theta : \mathcal{X} \to \mathcal{Y}$ – an inherently interpretable model by design (e.g. a linear model or a shallow decision tree),

3. **Instance-wise weight estimation model** $h_\phi : \mathcal{X} \times \mathcal{X} \times \mathcal{Y} \to [0, 1]$ – a function that outputs the instance-wise weights to fit the locally interpretable model. It uses concatenation of a probe feature, a training feature, and a corresponding black-box model prediction on the training feature as its inputs. It can be a complex machine learning model – e.g. here a deep neural network.

Our objective is to construct an accurate locally interpretable model $g_\theta$ such that the predictions made by it are similar to the predictions of the given black-box model $f^*$ – i.e. the locally interpretable model has high fidelity. We use a loss function, $\mathcal{L} : \mathcal{Y} \times \mathcal{Y} \to \mathbb{R}$ to quantify the fidelity of the locally interpretable model (e.g. mean absolute error, lower the better). In RL-LIM, the three necessary components of an RL framework are as follows: the state is the vector of input features, the action is the selection vector, and the reward is the fidelity which depends on the input features and the selection vector. The instance-wise weight estimator model is the agent that outputs the actions based on the state (input features). The environment is comprised of the input feature generating process, as well as the black-box model for the target task.

The representational capacity difference between the black-box model and the locally interpretable model is the bottleneck we aim to address. Ideally, to avoid underfitting, locally interpretable models should be learned with a minimal number of training instances that are most effective in capturing the model behavior. We propose an instance-wise weight estimation model $h_\phi$ to estimate the probabil-

ities of training instances that should be used for fitting the locally interpretable model. Integrating with the accurate locally interpretable modeling goal, we propose the following objective:

$$
\begin{aligned}
\min_{h_\phi} \quad & \mathbb{E}_{\mathbf{x}^p \sim P_X} \left[ \mathcal{L}(f^*(\mathbf{x}^p), g^*_{\theta(\mathbf{x}^p)}(\mathbf{x}^p)) \right] + \lambda \mathbb{E}_{\mathbf{x}^p, \mathbf{x} \sim P_X} \left[ h_\phi(\mathbf{x}^p, \mathbf{x}, f^*(\mathbf{x})) \right] \\
\text{s.t.} \quad & g^*_{\theta(\mathbf{x}^p)} = \arg\min_{g_\theta} \mathbb{E}_{\mathbf{x} \sim P_X} \left[ h_\phi(\mathbf{x}^p, \mathbf{x}, f^*(\mathbf{x})) \times \mathcal{L}_g(f^*(\mathbf{x}), g_\theta(\mathbf{x})) \right]
\end{aligned}
\tag{1}
$$

where $\lambda \geq 0$ is a hyper-parameter that controls the number of training instances used to fit the locally interpretable model (we study the impact of performance on $\lambda$ in Section 4.2), and $h_\phi(\mathbf{x}^p, \mathbf{x}, f^*(\mathbf{x}))$ represents the instance-wise weight for each training pair $(\mathbf{x}, f^*(\mathbf{x}))$ for the probe data $\mathbf{x}^p$. $\mathcal{L}_g$ is the loss function to fit the locally interpretable model, for which we use the mean squared error between predicted values for regression and logits for classification. $\phi$ and $\theta$ are the trainable parameters, whereas $f^*$ (the pre-trained black-box model) is fixed.

The first term in the objective function $\mathbb{E}_{\mathbf{x}^p \sim P_X} \left[ \mathcal{L}(f^*(\mathbf{x}^p), g^*_{\theta(\mathbf{x}^p)}(\mathbf{x}^p)) \right]$ represents the local prediction differences between black-box model and locally interpretable model (referred to as fidelity metric). The second term in the objective function $\mathbb{E}_{\mathbf{x}^p, \mathbf{x} \sim P_X} \left[ h_\phi(\mathbf{x}^p, \mathbf{x}, f^*(\mathbf{x})) \right]$ represents the expected number of selected training points to fit the locally interpretable model. Lastly, the constraint ensures that the locally interpretable model is derived from weighted loss function, where weights are the output of the instance-wise weight estimator $h_\phi$. Our formulation does not assume any constraint on $g_\theta$ – it could be any inherently interpretable model suitable for the data type of interest. Next, we describe how Eq. (1) can be efficiently addressed with reinforcement learning.

### 3.1 TRAINING AND INFERENCE

The RL-LIM method, shown in Fig. 1, can be thought of as encompassing 5 stages:

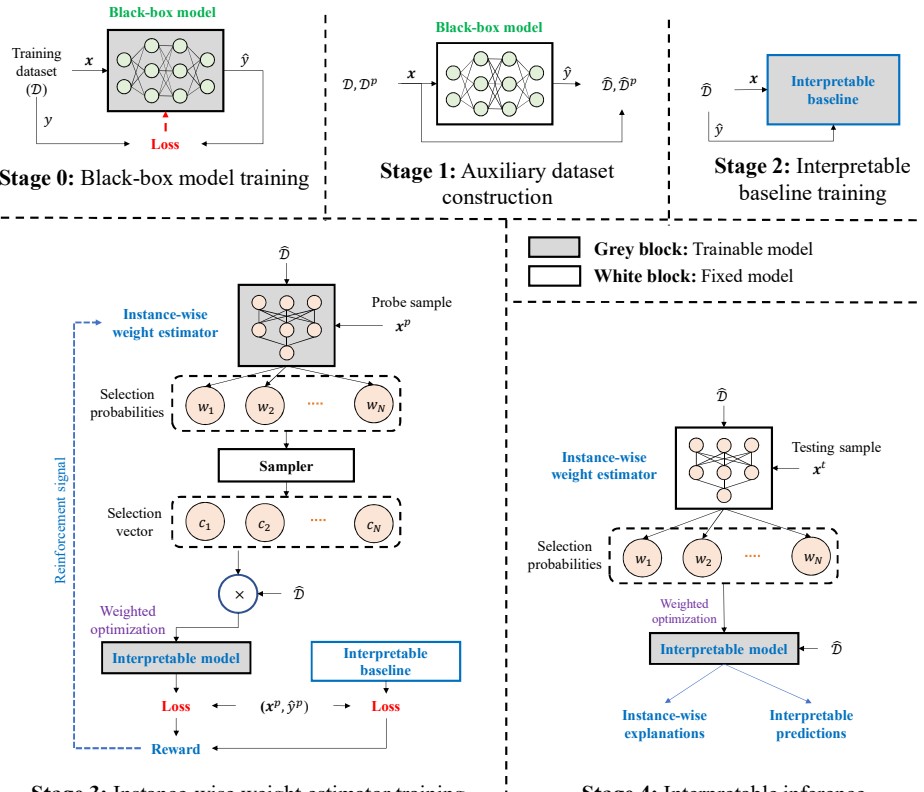

Figure 1: The proposed RL-LIM method. White blocks represent fixed (not learnable) models, and grey blocks represent learnable (trainable) models. **Stage 0:** Black-box model training. **Stage 1:** Auxiliary dataset construction. **Stage 2:** Interpretable baseline training. **Stage 3:** Instance-wise weight estimator training. **Stage 4:** Interpretable inference.

- **Stage 0 – Black-box model training**: This stage is the preliminary stage for RL-LIM. Given the training set $\mathcal{D}$, the black-box model $f$ is trained to minimize a loss function ($\mathcal{L}_f$) (e.g. mean squared error for regression or cross-entropy for classification), i.e., $f^* = \arg\min_f \frac{1}{N} \sum_{i=1}^{N} \mathcal{L}_f(f(\mathbf{x}_i), y_i)$. If the pre-trained black-box model is already saved, we can skip this stage and retrieve the given pre-trained black-box model to $f^*$.

- **Stage 1 – Auxiliary dataset construction**: Using the pre-trained black-box model $f^*$, we create auxiliary training and probe datasets, as $\hat{\mathcal{D}} = \{(\mathbf{x}_i, \hat{y}_i), i = 1, ..., N\}$ (where $\hat{y}_i = f^*(\mathbf{x}_i)$) and $\hat{\mathcal{D}}^p = \{(\mathbf{x}_j^p, \hat{y}_j^p), j = 1, ..., M\}$ (where $\hat{y}_j^p = f^*(\mathbf{x}_j^p)$), respectively. These auxiliary datasets ($\hat{\mathcal{D}}$, $\hat{\mathcal{D}}^p$) are used for instance-wise weight estimation models and locally interpretable model training.

- **Stage 2 – Interpretable baseline training**: To improve the stability of the instance-wise weight estimator training, a baseline model is observed to be beneficial. As the baseline model $g_b : \mathcal{X} \to \mathcal{Y}$, we use a globally interpretable model (such as a linear model or shallow decision tree) optimized to replicate the predictions of the black-box model: $g_b^* = \arg\min_g \frac{1}{N} \sum_{i=1}^{N} \mathcal{L}(g(\mathbf{x}_i), \hat{y}_i)$.

- **Stage 3 – Instance-wise weight estimator training**: We train an instance-wise weight estimator using the auxiliary datasets ($\hat{\mathcal{D}}$, $\hat{\mathcal{D}}^p$). To encourage exploration, we consider probabilistic selection, with a sampler block that is based on the output of the instance-wise weight estimator – $h_\phi(\mathbf{x}_j^p, \mathbf{x}_i, \hat{y}_i)$ represents the probability that $(\mathbf{x}_i, \hat{y}_i)$ is selected to train locally interpretable model for the probe instance $\mathbf{x}_j^p$. Let the binary vector $\mathbf{c}(\mathbf{x}_j^p) \in \{0,1\}^N$ represent the selection operation, such that $(\mathbf{x}_i, \hat{y}_i)$ is selected for training locally interpretable model for $\mathbf{x}_j^p$ when $c_i(\mathbf{x}_j^p) = 1$. Correspondingly, $\rho_\phi(\mathbf{x}^p)$ is the probability mass function for $\mathbf{c}(\mathbf{x}_j^p)$ given $h_\phi(\cdot)$:

$$\rho_\phi(\mathbf{x}_j^p, \mathbf{c}(\mathbf{x}_j^p)) = \prod_{i=1}^{N} \left[ h_\phi(\mathbf{x}_j^p, \mathbf{x}_i, f^*(\mathbf{x}_i))^{c_i(\mathbf{x}_j^p)} \cdot (1 - h_\phi(\mathbf{x}_j^p, \mathbf{x}_i, f^*(\mathbf{x}_i)))^{1-c_i(\mathbf{x}_j^p)} \right]$$

As the original form of the optimization problem in Eq. (1) is intractable due to the expectation operations, we employ approximations:

  - The sample mean is used as an approximation of the first term of the objective function as $\frac{1}{M} \sum_{j=1}^{M} \mathcal{L}(f^*(\mathbf{x}_j^p), g_{\theta(\mathbf{x}_j^p)}^*(\mathbf{x}_j^p))$.

  - The second term of the objective, which represents the average selection probability, is approximated as the number of selected instances (divided by $N$) to have $||\mathbf{c}(\mathbf{x}_j^p)||_1 = \frac{1}{N} \sum_{i=1}^{N} |c_i(\mathbf{x}_j^p)|$.

  - The constraint term is approximated using the sample mean of the training loss as $g_{\theta(\mathbf{x}_j^p)}^* = \arg\min_{g_\theta} \frac{1}{N} \sum_{i=1}^{N} \left[ c_i(\mathbf{x}_j^p) \cdot \mathcal{L}_g(f^*(\mathbf{x}_i), g_\theta(\mathbf{x}_i)) \right]$.

The sampler block yields a non-differential objective, and we cannot train the instance-wise weight estimator using conventional gradient descent-based optimization. There are approximations such as training in expectation (Raffel et al., 2017) or Gumbel-softmax (Jang et al., 2016). Instead, motivated by its many successful applications (Ranzato et al., 2015; Zaremba & Sutskever, 2015; Zhang & Lapata, 2017), we use REINFORCE algorithm (Williams, 1992) such that the selection action is rewarded by the performance of its impact. The loss function for the instance-wise weight estimator $l(\phi)$ is expressed as:

$$l(\phi) = \mathbb{E}_{\mathbf{x}_j^p \sim P_X} \left[ \mathbb{E}_{\mathbf{c}(\mathbf{x}_j^p) \sim \rho_\phi(\mathbf{x}_j^p, \cdot)} \left[ \mathcal{L}(f^*(\mathbf{x}_j^p), g_{\theta(\mathbf{x}_j^p)}^*(\mathbf{x}_j^p)) + \lambda ||\mathbf{c}(\mathbf{x}_j^p)||_1 \right] \right]$$

To apply the REINFORCE algorithm, we directly compute the gradient $\nabla_\phi \hat{l}(\phi)$ as:

$$\nabla_\phi \hat{l}(\phi) = \mathbb{E}_{\mathbf{x}_j^p \sim P_X} \left[ \mathbb{E}_{\mathbf{c}(\mathbf{x}_j^p) \sim \rho_\phi(\mathbf{x}_j^p, \cdot)} \left[ \mathcal{L}(f^*(\mathbf{x}_j^p), g_{\theta(\mathbf{x}_j^p)}^*(\mathbf{x}_j^p)) + \lambda ||\mathbf{c}(\mathbf{x}_j^p)||_1 \right] \nabla_\phi \log \rho_\phi(\mathbf{x}_j^p, \mathbf{c}(\mathbf{x}_j^p)) \right]$$

Using the gradient $\nabla_\phi \hat{l}(\phi)$, we employ the following steps iteratively to update the parameters of the instance-wise weight estimator $\phi$:

1. Estimate instance-wise weights $w_i(\mathbf{x}_j^p) = h_\phi(\mathbf{x}_j^p, \mathbf{x}_i, \hat{y}_i)$ and instance-wise selection vector $c_i(\mathbf{x}_j^p) \sim \text{Ber}(w_i(\mathbf{x}_j^p))$ for each training and probe instance in a mini-batch.

2. Optimize the locally interpretable model with the selection vector for each probe instance:

$$g_{\theta(\mathbf{x}_j^p)}^* = \arg\min_{g_\theta} \sum_{i=1}^N \left[ c_i(\mathbf{x}_j^p) \cdot \mathcal{L}_g(f^*(\mathbf{x}_i), g_\theta(\mathbf{x}_i)) \right]$$

3. Update the instance-wise weight estimation model parameter $\phi$:

$$\phi \leftarrow \phi - \frac{\alpha}{M} \sum_{j=1}^M \left[ \mathcal{L}(f^*(\mathbf{x}_j^p), g_{\theta(\mathbf{x}_j^p)}^*(\mathbf{x}_j^p)) - \mathcal{L}_b(\mathbf{x}_j^p) + \lambda ||\mathbf{c}(\mathbf{x}_j^p)||_1 \right] \cdot \nabla_\phi \log \rho_\phi(\mathbf{x}_j^p, \mathbf{c}(\mathbf{x}_j^p))$$

where $\alpha > 0$ is a learning rate and $\mathcal{L}_b(\mathbf{x}_j^p) = \mathcal{L}(f^*(\mathbf{x}_j^p), g_b^*(\mathbf{x}_j^p))$ is the baseline loss against which we benchmark the performance improvement. We repeat the steps above until convergence.

- **Stage 4 – Interpretable inference**: Unlike when training, we use a *fixed* instance-wise weight estimator (without the sampler and interpretable baseline) and merely fit the locally interpretable model at inference. Given the test instance $\mathbf{x}^t$, we obtain the selection probabilities from the instance-wise weight estimator, and using these as the weights, we fit the locally interpretable model via weighted optimization. The outputs of the trained interpretable model are the instance-wise predictions and the corresponding explanations (e.g., local dynamics of the black-box model predictions at $\mathbf{x}^t$ given by the coefficients of the fitted linear model).

## 3.2 COMPUTATIONAL COST

In this subsection, we analyze the computational cost of RL-LIM for training and inference. As a representative and commonly used example, we assume linear regression as the locally interpretable model, which has a computational complexity of $\mathcal{O}(d^2N) + \mathcal{O}(d^3)$ to fit, where $d$ is the number of features and $N$ is the number of training instances. When $N >> d$ (which is often the case in practice), the training computational complexity is approximated as $\mathcal{O}(d^2N)$ (Tan, 2018).

**Training:** Given a pre-trained black-box model, Stage 1 involves running inference $N$ times and the total complexity depends on the complexity of the black-box model. Unless the black-box model is very complex, the computational complexity of Stage 1 becomes much smaller than Stage 3. Stage 2 has negligible computational overhead. At Stage 3, we iteratively train the instance-wise weight estimator and fit the locally interpretable model from scratch using weighted optimization. Therefore, the computational complexity is $\mathcal{O}(d^2NN_I)$ where $N_I$ is the number of iterations in Stage 3 (typically $N_I < 10,000$ until convergence). Thus, the training complexity scales roughly linearly with the number of training instances.

**Interpretable inference:** To infer with the locally interpretable model, we need to fit the locally interpretable model after obtaining the instance-wise weights from the trained instance-wise weight estimator. Thus, for each testing instance, the computational complexity is $\mathcal{O}(d^2N)$.[1]

For instance, on a single NVIDIA V100 GPU, on Facebook Comment dataset (consisting $\sim 600,000$ samples), RL-LIM yields a training time of less than 5 hours (including Stage 1, 2 and 3) and an interpretable inference time of less than 10 seconds per a testing instance. On the other hand, LIME results in much longer interpretable inference time (around 30 seconds per a testing instance) due to acquiring a large number of black-box model predictions for the inputs perturbations, whereas SILO and MAPLE are similar to RL-LIM.

## 4 EXPERIMENTS

We compare RL-LIM to multiple benchmarks on 3 synthetic datasets and 5 UCI public datasets.

**Datasets:** The 3 public datasets for regression problems are: (1) Blog Feedback, (2) Facebook Comment, (3) News Popularity; the other 2 public datasets for classification problems are: (4) Adult Income, (5) Weather. Details of the data descriptions can be found in the hyper-links of each dataset (colored in blue). Data statistics can be found in Table 3 in Appendix A. In this section, we mainly focus on the tabular datasets because the local dynamics are more important and useful to explain for them; however, RL-LIM method can be generalized to other data types in a straightforward way.

---

[1] A subset of the training dataset can be used to reduce complexity (with decreased fidelity).

**Black-box models:** We focus on approximating black-box models that are shown to yield competitive performance on the target tasks: 3 tree-based ensemble methods (1) XGBoost (Chen & Guestrin, 2016), (2) LightGBM (Ke et al., 2017), (3) Random Forests (RF) (Breiman, 2001); and deep neural networks (4) Multi-layer Perceptron (MLP). Also, we use (5) Ridge Regression (RR) and (6) Regression Tree (RT) (for regression) and (7) Logistic Regression (LR) and (8) Decision Tree (DT) (for classification) as globally interpretable models to benchmark.[2] We focus on two types of locally interpretable models: (1) Ridge regression, (2) Shallow regression tree (with a max depth of 3). We report the performance with ridge regression for regression and with shallow regression tree for classification in this section. The results of the other two combinations (with ridge regression for classification and with shallow regression tree for regression) are described in Appendix E.

**Comparisons to previous work:** We compare the performance of RL-LIM with three competing methods: (1) Local Interpretable Model-agnostic Explanations (LIME) (Ribeiro et al., 2016), (2) Supervised Local modeling methods (SILO) (Bloniarz et al., 2016), (3) Model Agnostic Supervised Local Explanations (MAPLE) (Plumb et al., 2018).

**Performance metrics:** To evaluate the performance of locally interpretable models using real-world datasets, we quantify the overall prediction performance and its fidelity. We assume a disjoint testing dataset $\mathcal{D}^t = \{(\mathbf{x}_k^t, y_k^t)\}_{k=1}^L$ for evaluation. For the overall prediction performance, we compare the predictions of the locally interpretable models with the ground-truth labels. We use Mean Absolute Error (MAE) for regression and Average Precision Recall (APR) for classification. For fidelity, we compare the outputs (predicted values for regression and logits for classification) of the locally interpretable models and of the black-box model. We consider two metrics: $R^2$ score (Legates & McCabe, 1999) and Local MAE (LMAE). The details of the metrics are described in Appendix C.

**Implementation details:** We implement instance-wise weight estimator using a multi-layer perceptron with tanh activation. The number of hidden units and layers are optimized by the cross-validation. In most cases, 5-layer perceptron with 100 hidden units performs reasonably-well across all datasets. All features are normalized to be between zero and one, using standard minmax scaler. Categorical variables are transformed using one-hot encoding.

## 4.1 EXPERIMENTS ON SYNTHETIC DATASETS – RECOVERING LOCAL DYNAMICS

On real-world datasets it is challenging to directly evaluate the explanation quality of the locally interpretable models due to the absence of ground-truth explanations. Thus we initially focus on synthetic datasets (with known ground-truth explanations) to directly evaluate how well the locally interpretable models can recover the underlying local dynamics. We construct three synthetic datasets such that the 11-dimensional input features $\mathbf{X}$ are sampled from $\mathcal{N}(0, I)$ and $Y$ are:

1. Syn1: $Y = X_1 + 2X_2$ if $X_{10} < 0$ and $Y = X_3 + 2X_4$ if $X_{10} \geq 0$
2. Syn2: $Y = X_1 + 2X_2$ if $X_{10} + e^{X_{11}} < 1$ and $Y = X_3 + 2X_4$ if $X_{10} + e^{X_{11}} \geq 1$
3. Syn3: $Y = X_1 + 2X_2$ if $X_{10} + X_{11}^3 < 0$ and $Y = X_3 + 2X_4$ if $X_{10} + X_{11}^3 \geq 0$

All three datasets have different local dynamics in different input regimes. We directly use the ground truth function as the black-box model and focus on how well locally interpretable modeling can capture the local dynamics. We evaluate the performance of capturing local dynamics using Absolute Weight Difference (AWD): $\text{AWD} = ||\mathbf{w} - \hat{\mathbf{w}}||$, where $\mathbf{w}$ is the ground truth coefficients to generate $Y$ and $\hat{\mathbf{w}}$ is the derived coefficient from the locally interpretable models. We use the estimated coefficients of the ridge regression as the derived local dynamics ($\hat{\mathbf{w}}$).

As shown in Fig. 2, RL-LIM significantly outperforms other benchmarks in discovering the local dynamics on all three datasets and in different regimes. RL-LIM can actively learn the linear and non-linear decision boundaries for the local dynamics. Note that LIME completely fails to recover the local dynamics as it uses the Euclidean distance uniformly for all features and cannot distinguish the special properties of the features that alter the local dynamics. SILO and MAPLE only use the predictions to discover the local dynamics; thus, it is hard to discover the decision boundary that depends on the other variables which are independent to the predictions. Fig. 5 in Appendix D shows the learning curves of RL-LIM demonstrating the efficiency of reinforcement learning.

---

[2]We use python packages (including Sklearn and Tensorflow) to implement those predictive models and the details can be found in the hyper-links (colored in blue) of each model and Appendix B.

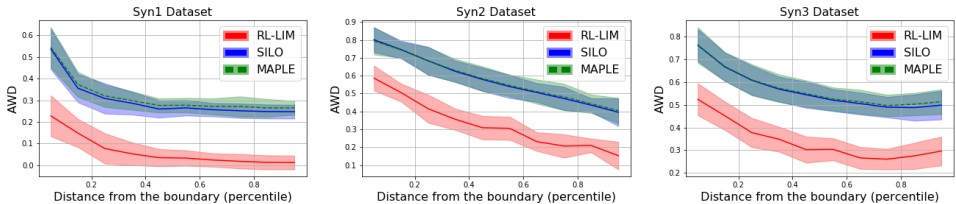

Figure 2: Synthetic dataset results. Mean absolute weight difference (AWD) with 95% confidence intervals (of 10 independent runs) on three synthetic datasets. X-axis: Distance from the boundary where the local dynamics change, such as $X_{10} = 0$ for Syn1 (in percentile), Y-axis: AWD (the lower, the better). We exclude LIME in these graphs due to its poor performance in terms of AWD (it is higher than 1.6 in all distance regimes for all three synthetic datasets).

## 4.2 THE EFFECT OF THE NUMBER OF SELECTED SAMPLES ON FIDELITY

In RL-LIM, optimal distillation is enabled by using a small subset of training instances to fit the low-capacity locally interpretable model. The number of selected instances is controlled by $\lambda$ in our method – if $\lambda$ is high/low, RL-LIM penalizes more/less on the number of selected instances; thus, less/more instances are selected to construct the locally interpretable model.

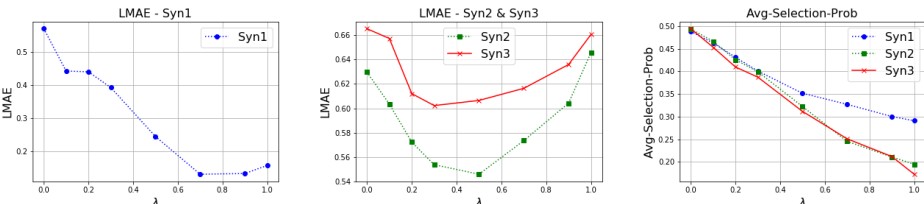

Figure 3: Fidelity & average selection probability of training instances as a function of the number of selected samples on three synthetic datasets. X-axis: $\lambda$, Y-axis: LMAE and average selection probability of training instances. LMAE is Local MAE – lower is better.

We analyze the efficacy of $\lambda$ in controlling the likelihood of selection and the dependency of fidelity on $\lambda$. We expect that if we select a too small/large number of training instances, the locally interpretable model will overfit/underfit which negatively affects the fidelity in both cases. Fig. 3 shows that there is a clear relationship between $\lambda$ and the local fidelity. If $\lambda$ is too large, RL-LIM selects too small number of instances; thus, the fitted locally interpretable model is less accurate (due to overfitting). On the other hand, if $\lambda$ is too small, RL-LIM selects too large number of instances and deteriorates fidelity (due to underfitting). To achieve the optimal $\lambda$, we conduct cross-validation experiments and select $\lambda$ which achieves the best validation fidelity (e.g. $\lambda = 0.5$ in Syn2). Fig. 3 shows the average selection probability of the training instances for each $\lambda$. As $\lambda$ increases, the average selection probabilities monotonically decrease due to the higher penalty on the number of selected training instances. Note that even using a small portion of training instances, RL-LIM can accurately distill the predictions of black-box models into locally interpretable models which is crucial to understand and interpret the predictions using the most relevant training instances.

## 4.3 EXPERIMENTS ON REAL DATASETS – OVERALL PERFORMANCE AND FIDELITY

On multiple real datasets, we evaluate the overall prediction performance and fidelity. For the regression and classification problems, we use ridge regression and shallow regression trees as the locally interpretable model. More results can be found in Appendix E.

As can be seen in Table 1, the performance of globally interpretable ridge regression (trained on the entire dataset from the scratch) is much worse than other complex non-linear models, implying that modeling non-linear relationships between the features and the labels is important towards high prediction performance. For other locally interpretable modeling methods (LIME, SILO, MAPLE), the performance is far worse than the original black-box model, showing that they fail at efficiently

| Datasets | Models | XGBoost | | LightGBM | | MLP | | RF | |
|---|---|---|---|---|---|---|---|---|---|
| (RR-MAE) | Metrics | MAE | $R^2$ | MAE | $R^2$ | MAE | $R^2$ | MAE | $R^2$ |
| **Blog** (8.420) | Original | 5.131 | 1.0 | 4.965 | 1.0 | 4.893 | 1.0 | 5.203 | 1.0 |
| | RL-LIM | **5.289** | **.8679** | **4.971** | **.9069** | **4.994** | **.7177** | **4.993** | **.8573** |
| | LIME | 9.421 | .3440 | 10.243 | .3019 | 10.936 | -.2723 | 19.222 | -.2143 |
| | SILO | 6.261 | .0005 | 6.040 | .2839 | 5.413 | .4274 | 6.610 | .4500 |
| | MAPLE | 5.307 | .8248 | 4.981 | .8972 | 5.012 | .5624 | 5.058 | .8471 |
| **Facebook** (24.64) | Original | 24.18 | 1.0 | 20.22 | 1.0 | 18.36 | 1.0 | 30.09 | 1.0 |
| | RL-LIM | **22.92** | **.7071** | **24.84** | **.4268** | **20.23** | **.5495** | **22.65** | **.4360** |
| | LIME | 35.20 | .2205 | 38.19 | .2159 | 38.82 | .2463 | 51.77 | .1797 |
| | SILO | 31.41 | -.4305 | 39.10 | -1.994 | 22.35 | .3307 | 42.05 | -.7929 |
| | MAPLE | 23.28 | .6803 | 41.86 | -3.233 | 24.77 | -.1721 | 44.75 | -1.078 |
| **News** (2989) | Original | 2995 | 1.0 | 3140 | 1.0 | 2255 | 1.0 | 3378 | 1.0 |
| | RL-LIM | **2958** | **.7534** | **2957** | **.5936** | 2260 | **.9761** | **2396** | **.6523** |
| | LIME | 5141 | -.2467 | 6301 | -2.008 | 2289 | .5030 | 9435 | -7.477 |
| | SILO | 3069 | .4547 | 3006 | .4025 | **2257** | .9617 | 3251 | .3816 |
| | MAPLE | 2967 | .7010 | 3005 | .3963 | 2259 | .9534 | 3060 | .5901 |

Table 1: Real-world regression dataset results. Overall prediction performance (metric: MAE, lower is better) and fidelity (metric: $R^2$ score, higher is better) on regression problems with ridge regression as the locally interpretable model. 'Original' is the performance of the original black-box model that the models are approximating. MAE of global ridge regression (RR) can be found below the data name. Red represents performance that is worse than global ridge regression and the negative $R^2$ scores. **Bold** represents the best results.

distilling the non-linear black-box models. In some cases (especially on the Facebook dataset), the performance of the benchmarks is even worse than the performance of global ridge regression (highlighted in red), questioning the value of using these locally interpretable models instead of globally interpretable ridge regression.

In contrast, RL-LIM achieves similar overall prediction performance to the black-box models and significantly outperforms global ridge regression. Table 1 also compares the fidelity in terms of $R^2$ score for regression using ridge regression as the locally interpretable model (LMAE results can be found in Appendix E.3). We observe that $R^2$ scores for some cases (especially on Facebook dataset and LIME) are negative which represent that the outputs of the locally interpretable models are even worse than the constant mean value estimator. On the other hand, RL-LIM achieves higher and positive $R^2$ values consistently for all datasets and black-box models than other benchmarks.

Table 2 shows a similar analysis for classification using shallow regression trees (with max depth of 3) as the locally interpretable model (Regression trees are used to model logit outputs for classification.). The overall prediction performance of four black-box models are significantly better than the globally interpretable decision tree which demonstrates the superior fitting by complex black-box models. Among the locally interpretable models, RL-LIM achieves the best APR and $R^2$ score for most cases, underlining its strength in distilling the predictions of the black-box model accurately. In some cases, the benchmarks (especially for LIME) achieve lower overall prediction performance than the globally interpretable decision tree (highlighted in red). The overall prediction performance and fidelity metrics of all locally interpretable models seem better for classification problems than regression problems. We expect that the predictions of black-box models are mostly highly confident, i.e. located near 0 or 1; thus, locally interpretable models can easily distill the predictions of the black-box models for classification than regression.

## 4.4 QUALITATIVE ANALYSES – INTERPRETATIONS OF RL-LIM ON ADULT INCOME DATASET

In this subsection, we qualitatively analyze the local explanations provided by RL-LIM on the Adult Income dataset. Although RL-LIM is able to provide local explanations for each individual separately, we analyze its explanations in subgroup granularity for better visualization and understanding. Fig. 4 represents the feature importance (derived by RL-LIM as the local explanations) for

| Datasets | Models | XGBoost | | LightGBM | | MLP | | RF | |
|---|---|---|---|---|---|---|---|---|---|
| (DT-APR) | Metrics | APR | $R^2$ | APR | $R^2$ | APR | $R^2$ | APR | $R^2$ |
| **Adult** (.6388) | Original | .8096 | 1.0 | .8254 | 1.0 | .7678 | 1.0 | .7621 | 1.0 |
| | RL-LIM | **.8011** | **.9889** | **.8114** | **.9602** | .7710 | .9451 | **.7881** | **.8788** |
| | LIME | .6211 | .5009 | .6031 | .3798 | .4270 | .2511 | .6166 | .3833 |
| | SILO | .8001 | .9869 | .8107 | .9583 | .7708 | **.9470** | .7833 | .8548 |
| | MAPLE | .7928 | .9794 | .8034 | .9405 | **.7719** | .9410 | .7861 | .8622 |
| **Weather** (.5838) | Original | .7133 | 1.0 | .7299 | 1.0 | .7205 | 1.0 | .7274 | 1.0 |
| | RL-LIM | **.7071** | **.9734** | **.7118** | **.9601** | **.7099** | **.9124** | **.7102** | **.9008** |
| | LIME | .6179 | .7783 | .6159 | .6913 | .5651 | .3417 | .6209 | .3534 |
| | SILO | .6991 | .9680 | .7052 | .9452 | .6997 | .8864 | .7042 | .8398 |
| | MAPLE | .6973 | .9675 | .7056 | .9446 | .6983 | .8856 | .6983 | .8856 |

Table 2: Real-world classification dataset results. Overall prediction performance (metric: APR, higher is better) and fidelity (metric: $R^2$ score, higher is better) on classification problems with shallow regression tree as the locally interpretable model. 'Original' is the performance of the original black-box model that the models are approximating. APR of global decision tree (DT) can be found below the data name.

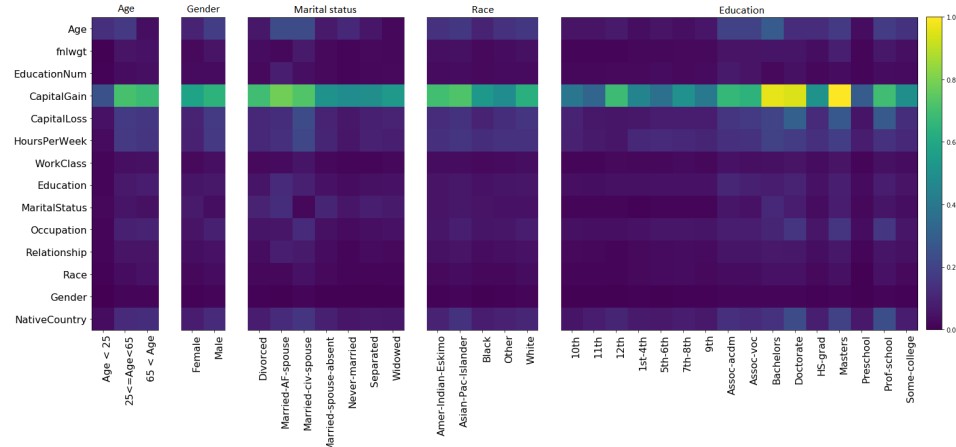

Figure 4: Qualitative interpretability results. The analyses of feature importance (derived by RL-LIM) for 5 types of subgroups in Adult Income dataset: (a) Age, (b) Gender, (c) Marital status, (d) Race, (e) Education. The color represents the feature importance for each subgroup.

five subgroups in predicting the annual income using XGBoost as the black-box model. We use ridge regression as the locally interpretable model and the absolute value of fitted coefficients as the estimated feature importance. As can be observed in Fig. 4, for age subgroups, capital gain seems much more important for mature people (older than 25) than young people (younger than 25). For education subgroups, capital gain/loss, occupation, and native countries are more critical for highly-educated people (Doctorate, Prof-school, and Masters graduates) than the others. We do not discover notable biases of black-box models for gender, marital status, and race subgroups.

## 5 CONCLUSIONS

We propose a novel method for locally interpretable modeling of pre-trained black-box models. Our proposed method employs reinforcement learning to select a small number of valuable instances and use them to train a low-capacity locally interpretable model. The selection mechanism is guided with a reward obtained from the similarity of predictions of the locally interpretable model and the black-box model. Our approach near-matches the performance of black-box models and significantly outperforms alternative techniques in terms of overall prediction performance and fidelity metrics consistently across various datasets and black-box models.

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

## A  DATA STATISTICS

| Problem | Data Name | # of samples | # of features | Label distribution |
|---|---|---|---|---|
| Regression | Blog | 60,021 | 280 | 6.6 (0-0-22) |
| | Facebook | 603,713 | 54 | 7.2 (0-0-30) |
| | News | 39,644 | 59 | 3395.4 (584-1400-10800) |
| Classification | Adult | 48,842 | 108 | 11,687 (23.9%) |
| | Weather | 112,925 | 61 | 25,019 (22.2%) |

Table 3: Data Statistics (# represents the number). Label distributions: # of positive labels (positive label ratio) for classification problem, Mean (5%-50%-95% percentiles) for regression problem.

## B  HYPER-PARAMETERS OF THE PREDICTIVE MODELS

In this paper, we use 8 different predictive models. For each predictive model, the corresponding hyper-parameters used in the experiments are as follows:

- **XGBoost:** booster - gbtree, max depth - 6, learning rate - 0.3, number of estimators - 1000, max depth - 6, reg alpha - 0
- **LightGBM:** booster - gbdt, max depth - None, learning rate - 0.1, number of estimators - 1000, min data in leaf - 20
- **Random Forests:** number of estimators - 1000, criterion - gini, max depth - None, warm start - False
- **Multi-layer Perceptron:** Number of layers - 4, hidden units - [feature dimensions, feature dimensions/2, feature dimensions/4, feature dimensions/8], activation function - relu, early stoping - True with patient 10, batch size - 256, maximum number of epochs - 200, optimizer - Adam
- **Ridge Regression:** alpha - 1
- **Regression Tree:** max depth - 3, criterion - gini
- **Logistic Regression:** solver - lbfgs, no regularization
- **Decision Tree:** max depth - 3, criterion - gini

We follow the default settings for the other hyper-parameters that are not mentioned here.

## C    PERFORMANCE METRICS

- **Mean Absolute Error (MAE):**

$$\text{MAE} = \mathbb{E}_{(\mathbf{x}^t, y^t) \sim \mathcal{P}} ||g^*_{\theta(\mathbf{x}^t)}(\mathbf{x}^t) - y^t)||_1 \simeq \frac{1}{L} \sum_{k=1}^{L} ||g^*_{\theta(\mathbf{x}^t_k)}(\mathbf{x}^t_k) - y^t_k||_1,$$

- **Local MAE (LMAE):**

$$\text{LMAE} = \mathbb{E}_{\mathbf{x}^t \sim \mathcal{P}_X} ||g^*_{\theta(\mathbf{x}^t)}(\mathbf{x}^t) - f^*(\mathbf{x}^t)||_1 \simeq \frac{1}{L} \sum_{k=1}^{L} ||g^*_{\theta(\mathbf{x}^t_k)}(\mathbf{x}^t_k) - f^*(\mathbf{x}^t_k))||_1,$$

- $R^2$ **score** (Legates & McCabe, 1999):

$$R^2 = 1 - \frac{\mathbb{E}_{\mathbf{x}^t \sim \mathcal{P}_X} ||f^*(\mathbf{x}^t) - g^*_{\theta(\mathbf{x}^t)}(\mathbf{x}^t)||_2^2}{\mathbb{E}_{\mathbf{x}^t \sim \mathcal{P}_X} ||f^*(\mathbf{x}^t) - \mathbb{E}_{\hat{\mathbf{x}}^t \sim \mathcal{P}_X}[f^*(\hat{\mathbf{x}}^t)]||_2^2} \simeq 1 - \frac{\frac{1}{L} \sum_{k=1}^{L} ||f^*(\mathbf{x}^t_k) - g^*_{\theta(\mathbf{x}^t_k)}(\mathbf{x}^t_k)||_2^2}{\frac{1}{L} \sum_{k=1}^{L} ||f^*(\mathbf{x}^t_k) - \frac{1}{L} \sum_{k=1}^{L}[f^*(\mathbf{x}^t_k)]||_2^2}.$$

If $R^2 = 1$, the predictions of the locally interpretable model perfectly match the predictions of the black-box model. On the other hand, if $R^2 = 0$, the locally interpretable model performs as similar as the constant mean value estimator. If $R^2 < 0$, the locally interpretable model performs worse than the constant mean value estimator.

## D    LEARNING CURVES OF RL-LIM

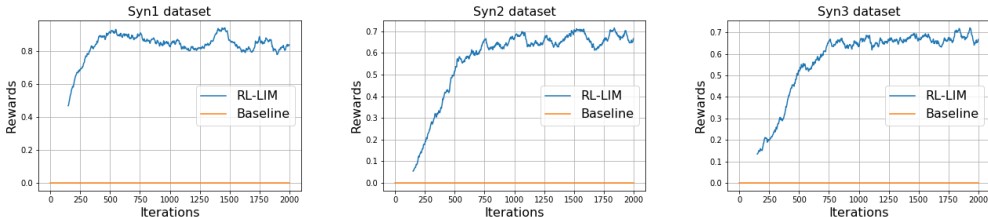

Figure 5: Learning curves of RL-LIM on three synthetic datasets. X-axis: The number of iterations on instance-wise weight estimator training, Y-axis: Rewards (LMAE of baseline (globally interpretable model) - LMAE of RL-LIM), higher the better.

# E  ADDITIONAL RESULTS

## E.1  REGRESSION WITH SHALLOW REGRESSION TREE AS THE LOCALLY INTERPRETABLE MODEL

| Datasets | Models | XGBoost | | LightGBM | | MLP | | RF | |
|---|---|---|---|---|---|---|---|---|---|
| (RT-MAE) | Metrics | MAE | $R^2$ | MAE | $R^2$ | MAE | $R^2$ | MAE | $R^2$ |
| **Blog** (5.955) | Original | 5.131 | 1.0 | 4.965 | 1.0 | 4.939 | 1.0 | 5.203 | 1.0 |
| | **RL-LIM** | **5.121** | **.8242** | **4.778** | **.8939** | **4.587** | **.6375** | **4.652** | **.8990** |
| | LIME | 11.80 | .2658 | 13.22 | .1483 | 7.396 | -.6201 | 19.61 | -.4116 |
| | SILO | 5.149 | .8035 | 4.818 | .8816 | 4.649 | .6177 | 4.715 | .8774 |
| | MAPLE | 5.329 | .7991 | 5.024 | .8660 | 4.609 | .6339 | 5.016 | .8201 |
| **Facebook** (22.28) | Original | 24.18 | 1.0 | 20.22 | 1.0 | 18.36 | 1.0 | 30.09 | 1.0 |
| | **RL-LIM** | **21.82** | **.9307** | **21.35** | **.9194** | **18.56** | **.8832** | **22.44** | **.7236** |
| | LIME | 36.69 | .3278 | 44.21 | .1809 | 40.85 | -.1513 | 51.70 | .2301 |
| | SILO | 22.42 | .8655 | 22.33 | .7235 | 19.57 | .8566 | 24.41 | .6917 |
| | MAPLE | 22.15 | .8824 | 23.43 | .8581 | 20.32 | .8035 | 27.12 | .3134 |
| **News** (3093) | Original | 2995 | 1.0 | 3140 | 1.0 | 2255 | 1.0 | 3378 | 1.0 |
| | **RL-LIM** | 2938 | **.9382** | **2504** | **.4104** | **2226** | **.9016** | **2431** | **.2768** |
| | LIME | 6272 | -.6267 | 7737 | -2.960 | 2390 | .0013 | 9637 | -7.075 |
| | SILO | **2910** | .1020 | 2854 | .3461 | 2274 | .8201 | 2874 | .2278 |
| | MAPLE | 2968 | .9288 | 2846 | .3631 | 2284 | .8021 | 2888 | .1872 |

Table 4: Real-world regression dataset results. Overall prediction performance (metric: MAE, lower is better) and fidelity (metric: $R^2$ score, higher is better) on regression problems with shallow regression tree as the locally interpretable model. 'Original' is the performance of the original black-box model that the models are approximating. MAE of global regression tree (RT) can be found below the data name. Red represents performance that is worse than global regression tree and the negative $R^2$ scores. **Bold** represents the best results.

| Datasets | Models | XGBoost | LightGBM | MLP | RF |
|---|---|---|---|---|---|
| Blog | **RL-LIM** | **.7530** | **1.358** | 1.273 | **1.413** |
| | LIME | 9.160 | 11.16 | 5.006 | 17.461 |
| | SILO | .8325 | 1.379 | **1.178** | 1.934 |
| | MAPLE | 1.029 | 1.598 | 1.359 | 2.158 |
| Facebook | **RL-LIM** | **7.240** | **6.867** | **5.596** | **15.77** |
| | LIME | 31.52 | 37.75 | 30.58 | 45.58 |
| | SILO | 8.459 | 9.149 | 6.997 | 18.63 |
| | MAPLE | 7.985 | 8.644 | 7.290 | 23.17 |
| News | **RL-LIM** | **389.0** | **1072** | **116.6** | **957.1** |
| | LIME | 4455 | 6243 | 504.0 | 9969 |
| | SILO | 496.7 | 1214 | 160.6 | 1175 |
| | MAPLE | 440.7 | 1201 | 163.6 | 1196 |

Table 5: Fidelity results (Metric: LMAE, lower the better) on regression problems with shallow regression tree as the locally interpretable model. **Bold** represents the best results.

## E.2 Classification with ridge regression as the locally interpretable model

| **Datasets** | Models | XGBoost | | LightGBM | | MLP | | RF | |
|---|---|---|---|---|---|---|---|---|---|
| (LR-APR) | Metrics | APR | $R^2$ | APR | $R^2$ | APR | $R^2$ | APR | $R^2$ |
| **Adult** (.7553) | Original | .8096 | 1.0 | .8254 | 1.0 | .7678 | 1.0 | .7621 | 1.0 |
| | **RL-LIM** | **.7977** | **.9871** | **.8039** | **.9439** | .7670 | **.9791** | .7977 | **.9217** |
| | LIME | .6803 | .7195 | .6805 | .6259 | .6957 | .8310 | .7057 | .6759 |
| | SILO | .7912 | .9750 | .7884 | .9301 | .7655 | .9778 | .7664 | .9140 |
| | MAPLE | .7947 | .9840 | .8011 | .9386 | **.7683** | .9636 | .7958 | .8961 |
| **Weather** (.7009) | Original | .7133 | 1.0 | .7299 | 1.0 | .7205 | 1.0 | .7274 | 1.0 |
| | **RL-LIM** | **.7140** | .9879 | **.7290** | **.9801** | .7212 | .9755 | **.7331** | **.9450** |
| | LIME | .6376 | .7898 | .6392 | .6873 | .6395 | .5321 | .6387 | .4513 |
| | SILO | .7134 | .9888 | .7281 | .9773 | **.7220** | .9797 | .7277 | .9024 |
| | MAPLE | .7134 | **.9897** | .7273 | .9778 | .7213 | .9702 | .7308 | .9323 |

Table 6: Real-world classification dataset results. Overall prediction performance (metric: APR, higher is better) and fidelity (metric: $R^2$ score, higher is better) on classification problems with ridge regression as the locally interpretable model. 'Original' is the performance of the original black-box model that the models are approximating. APR of global logistic regression (LR) can be found below the data name. Red represents the results that are worse than global logistic regression and the negative $R^2$ scores. **Bold** represents the best results.

## E.3 Regression with ridge regression as the locally interpretable model - Fidelity analysis in terms of Local MAE (LMAE)

| Datasets | Models | XGBoost | LightGBM | MLP | RF |
|---|---|---|---|---|---|
| Blog | **RL-LIM** | **.8679** | **1.135** | **1.432** | **1.651** |
| | LIME | 6.534 | 8.037 | 8.207 | 17.01 |
| | SILO | 2.220 | 3.046 | 2.393 | 3.909 |
| | MAPLE | .9690 | 1.416 | 1.550 | 1.984 |
| Facebook | **RL-LIM** | **6.394** | **21.29** | **8.217** | **33.64** |
| | LIME | 32.57 | 33.70 | 27.38 | 48.03 |
| | SILO | 19.51 | 30.07 | 11.52 | 40.14 |
| | MAPLE | 7.664 | 31.25 | 13.31 | 44.38 |
| News | **RL-LIM** | **436.9** | **1049** | **74.11** | **905.8** |
| | LIME | 3317 | 4766 | 327.4 | 8828 |
| | SILO | 657.2 | 1253 | 79.85 | 1345 |
| | MAPLE | 500.5 | 1261 | 88.19 | 1157 |

Table 7: Fidelity results (Metric: LMAE, lower the better) on regression problems with ridge regression as the locally interpretable model. **Bold** represents the best results.

### E.4 COMPARISON TO DIFFERENTIABLE WEIGHTING BASELINES

In this subsection, we additionally compare RL-LIM to two baselines that have differentiable objective for weighting: (1) straight-through estimator (STE) (Bengio et al., 2013), (2) Learning to Reweight (L2R) (Ren et al., 2018). The experimental setup is the same with the synthetic experiments in Section 4.1 and the performance metric is Average Weight Difference (AWD). As can be seen from Table 8, the first approach, STE, converges fast but to a suboptimal AWD, whereas L2R overfits to the fidelity metric and cannot guide weighting of the training samples properly, eventually yielding poor AWD.

| AWD | Syn1 | Syn2 | Syn3 |
|---|---|---|---|
| RL-LIM | **0.1562** | **0.3325** | **0.3920** |
| STE | 0.1717 | 0.3601 | 0.4307 |
| L2R | 0.7532 | 0.7283 | 0.7506 |

Table 8: Average Weight Differences (AWD) of RL-LIM, STE, and L2R on three synthetic datasets. **Bold** represents the best results.

The sampler block in RL-LIM makes the optimization problem of the RL-LIM non-differentiable. The main motivation of using the sampler in RL-LIM is to encourage exploration in order to systematically explore the extremely large action space. When we utilize straight-through estimator to make the entire process differentiable, the model converges faster but to a suboptimal solution due to the under-exploration of the action space.

The main difference between L2R and RL-LIM is that L2R learns weights of the training samples that are the same across all validation and testing samples; on the other hand, RL-LIM uses a single instance-wise weight estimator model to learn different weights of the training samples for each probe samples. If we apply L2R to this framework, we need to separately apply L2R for each probe sample. Therefore, there would be a high likelihood of overfitting to the probe dataset because it would only use one validation sample to learn the weights of the training samples.

### E.5 SAMPLE COMPLEXITY ANALYSIS

In this subsection, we report the sample complexity analysis results of RL-LIM, comparing to: (1) randomly selected subsamples (Random), (2) STE-based weighting (STE). We vary the number of training samples (from 200 to 2000) and compute the AWD on three synthetic datasets.

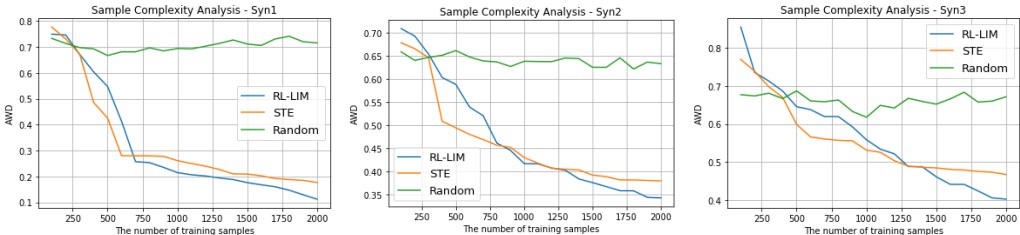

Figure 6: AWD performances in terms of the number of training samples used to train three models (RL-LIM, STE, Random) - Lower the better.

As can be seen in Fig. 6, with randomly selected subsamples, it only works well when we have an extremely small number of training samples due to a smaller chance of overfitting. For STE-based model, it converges fast; thus, it works better with smaller training samples; however, for larger training samples, it performs worse than RL-LIM due to under-exploration of the action space.

# F  ANALYSIS OF INSTANCE-WISE WEIGHTS DISTRIBUTIONS

In this section, we analyze the instance-wise weights of training samples visualizing the distributions of the instance-wise weights of training samples for the entire probe samples, and the dependence of the average instance-wise weights of training samples on the distance from the probe sample.

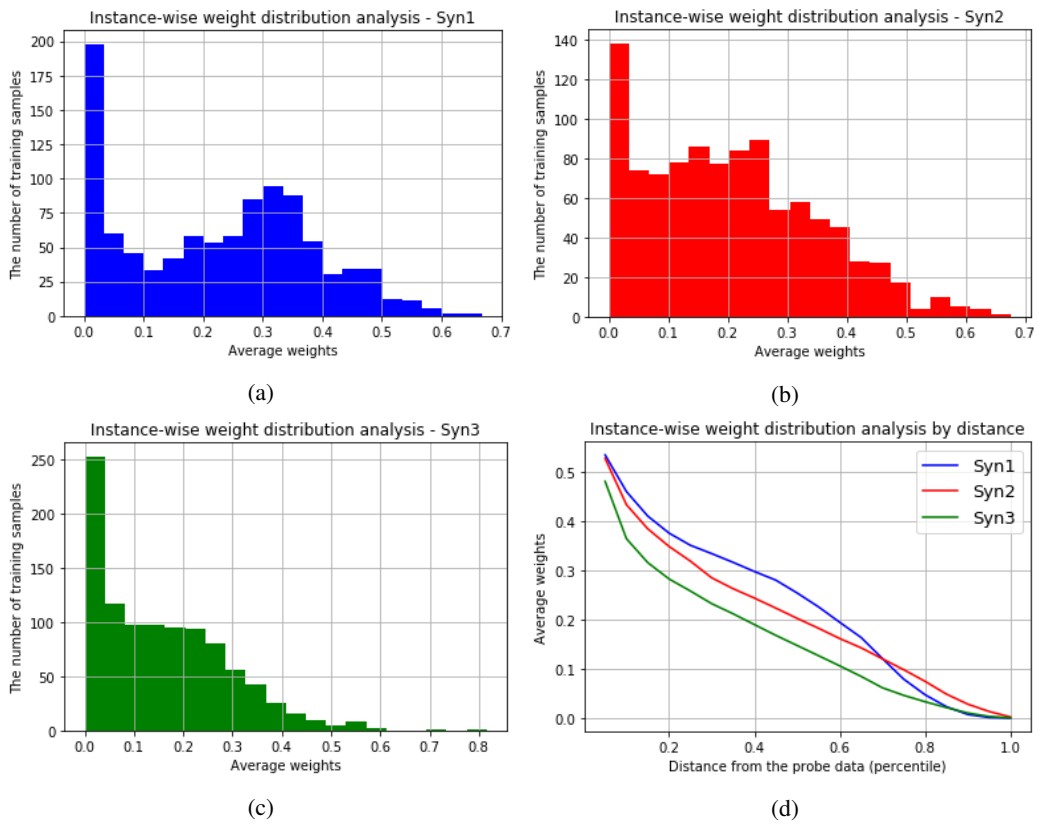

Figure 7: (a) Instance-wise weight distributions - Syn1, (b) Instance-wise weight distributions - Syn2, (c) Instance-wise weight distributions - Syn3, (d) Average instance-wise weights of training samples in terms of distance from the probe sample (percentile)

As can be seen in Fig. 7 (a)-(c), the instance-wise weights have quite skewed distribution. Some samples (e.g. with average instance-wise weights above 0.5) are much more critical to interpreting the probe sample than many others (e.g. average instance-wise weights below 0.1)

In Fig. 7 (d), there is a clear trend that the samples near the probe sample have higher average instance-wise weights, which shows that RL-LIM learns the meaningful distance metrics to measure the relevance while interpreting the probe samples. This trend is consistent across all three synthetic datasets.

