# OpenReview forum: "RL-LIM: Reinforcement Learning-based Locally Interpretable Modeling"
_ICLR.cc/2020/Conference — Reject_

### Official Review · AnonReviewer2 · 2019-10-23
**Official Blind Review #2**

**Rating:** 6

**Review:**

This paper studies the problem of interpreting predictions of blackbox models. In particular, they study local interpretable models, which are used to study interpretability at the level of one or a few data points. A key challenge is that in order for local models to be interpretable, they need to be simple in form and therefore lower in capacity (i.e. linear); thus, if they are trained on entire datasets they will underfit.

The aim of this work is to address this issue by learning how to select representative samples from a dataset for training local linear models to reproduce predictions of black-boxes. In particular, they propose to use RL to learn to weight instances from datasets; RL is required as they make hard (i.e. non-differentiable) decisions to select a subset of the dataset.

This work is closely related to Ren et al. 2018 [1], which proposes to meta-learn how to weight samples in a batch so as to maximize performance on a validation set. By analogy, in this paper, the samples are taken from the entire dataset (i.e. the dataset is subsampled), while the validation loss is effectively an imitation loss formed by treating the black-box model predictions as a target. The data subsampling operation introduces the added complication of non-differentiability. The contribution of this paper is thus the use of RL for meta-learning how to subsample a larger dataset in order to maximize some validation loss.

* Pros:
	* Considers an interesting dataset subsampling variant of the sample weighting meta-learning problem.
	* Novel application of meta-learning for improving locally-linear models.
	* Extensive quantitative evaluation shows that the method seems to perform better than baselines, though it might be that a differentiable approximation could do as well while being more sample efficient.
	* It is a nice result that the l1 penalty actually works well in reducing the number of samples chosen by the
	* I found the discussion and figures presented in 4.2 to be quite nice and informative.

* Cons:
	* Given the lack of a differentiable approximation baseline, I am not entirely convinced that the use of RL is absolutely necessary/optimal.
		* I.e. if the weighting function is actually high-entropy, randomly sampling a (large) batch and weighting it might work just as well.
	* Though there is discussion of the complexity of the overall method it would be nice to see a discussion and figures related to the sample efficiency of REINFORCE?
		* This would be strongest if given with a comparison to differentiable alternatives (mentioned above) as well.
		* This would help elucidate whether RL is optimal in this setting: fitting a linear model on more data might be cheaper learning to subsample with REINFORCE.
	* While the sample weighting function is fast at inference time, most of the overhead comes at training time. This function needs be updated in settings where the underlying dataset changes.
	* This is a minor issue, but this pushes the burden of interpretability further up to the black-box sample weighting function. While this interpretability problem is less critical, it still exists.

* Other comments/requests:
	* While the use of RL is certainly motivated in order to solve the problem in an unbiased way, it would be nice to see a comparison to a differentiable approximation as a baseline? A few ideas:
		* Randomly sample a (possibly large batch) and learn to weight it (closely related to the straight through estimator)
		* Randomly sample a batch and apply [1]
	* Would be nice to show the sizes of datasets and how many samples end up being used for different values of lambda.
	* Would be nice to understand which samples are chosen and why. This is probably tricky to analyze, but it would be interesting to see if certain samples are often chosen, or if the weighting distribution has an interesting shape (i.e. is low or high-entropy).

I’ve given a weak accept, conditioned on being provided more evidence regarding 1) comparisons to simple differentiable alternatives, 2) sample efficiency of the RL method, and 3) basic analysis of the weighting function.

[1] Learning to Reweight Examples for Robust Deep Learning. Mengye Ren, Wenyuan Zeng, Bin Yang, Raquel Urtasun. https://arxiv.org/abs/1803.09050


**Experience Assessment:**

I have read many papers in this area.

**Review Assessment: Checking Correctness Of Derivations And Theory:**

I assessed the sensibility of the derivations and theory.

**Review Assessment: Checking Correctness Of Experiments:**

I assessed the sensibility of the experiments.

**Review Assessment: Thoroughness In Paper Reading:**

I read the paper thoroughly.

---

> ### Author Response · Authors · 2019-11-15
> **RE: Official Blind Review #2 - 2**
>
> Answer 6: We appreciate this suggestion to improve our paper. We have included a new Section, Appendix F, on analysis of the instance-wise weights to build insights on the selected samples. We visualize the distributions of the instance-wise weights of training samples for the entire probe samples, and we show the dependence of the average instance-wise weights of training samples on the distance from the probe sample. As can be seen from these figures (Fig. 7 (a) - (c)) , the instance-wise weights have quite skewed distribution. Some samples (e.g. with average instance-wise weights above 0.5) are much more critical to interpreting the probe sample than many others (e.g. average instance-wise weights below 0.1)
>
> Furthermore, as can be seen in Fig. 7 (d) there is a clear trend that the samples near the probe sample have higher average instance-wise weights, which shows that RL-LIM learns the meaningful distance metrics to measure the relevance while interpreting the probe samples. This trend is consistent across all three synthetic datasets.
>
> We hope that we have fully addressed your questions and concerns. Please let us know if you have further comments.

---

> ### Author Response · Authors · 2019-11-15
> **RE: Official Blind Review #2 - 1**
>
> Thanks for all the positive comments on our paper - finding it novel, interesting, and informative with nice results.  See below for answers to questions as well as the additional requested experiments.
>
> Answer 1: We appreciate this suggestion, which has helped us to strengthen our motivation to use reinforcement learning.
>
> We have run two suggested differentiable baselines and added the results to the Appendix E.4 of the revised manuscript. In summary, our experiments on three synthetic datasets show that the first approach,  straight-through estimator (STE), converges fast but to a suboptimal Average Weight Difference (AWD) (0.1717, 0.3601, 0.4307), whereas Learning to Reweight (L2R) [1] overfits to the fidelity metric and yields a poor AWD (0.7532, 0.7283, 0.7506), in comparison to AWD of RL-LIM (0.1562, 0.3325, 0.3920), respectively (for Syn1, Syn2, Syn3 datasets).
>
> The sampler block in RL-LIM makes the optimization problem of the RL-LIM non-differentiable. The main motivation of using the sampler in RL-LIM is to encourage exploration in order to systematically explore the extremely large action space. When we utilize straight-through estimator to make the entire process differentiable, the model converges faster but to a suboptimal solution due to the under-exploration of the action space.
>
> The main difference between L2R and RL-LIM is that L2R learns weights of the training samples that are the same across all validation and testing samples; on the other hand, RL-LIM uses a single instance-wise weight estimator model to learn different weights of the training samples for each probe samples. If we apply L2R to this framework, we need to separately apply L2R for each probe sample. Therefore, there would be a high likelihood of overfitting to the probe dataset because it would only use one validation sample to learn the weights of the training samples.
>
> Answer 2: We have provided the sample complexity analysis for RL-LIM, comparing to two suggested benchmarks: (1) randomly selected subsamples, (2) STE-based model in the Appendix E.5. As can be seen in the results, with randomly selected subsamples, it only works well when we have an extremely small number of training samples due to a smaller chance of overfitting. For STE-based model, it converges fast; thus, it works better with smaller training samples; however, for larger training samples, it performs worse than RL-LIM due to under-exploration of the action space. For example, randomly selected subsamples works the best until 250 training samples, STE based model works the best from 250 to 700 training samples, and RL-LIM works the best with more than 700 training samples for Syn1 dataset (see Fig. 6 in the appendix).
>
> Answer 3: RL-LIM provides an interpretable model given a ‘fixed training dataset’, as a consequence of the problem setting we consider. For a different training dataset, RL-LIM needs to be retrained (the black-box model needs to be retrained as well). RL-LIM has computational overhead, but we show that even with a large-scale dataset such as Facebook Comment (consisting ∼ 600k samples), RL-LIM yields an entire training time of less than 5 hours, which is practically feasible in most settings.
>
> Answer 4: This is a valid perspective. Our objective is trying to optimize a locally-interpretable model (which is interpretable by design) to interpret the predictions of black-box model. To achieve this objective, we need a black-box sample weighting function; however, we mitigate the interpretability problem from the critical aspect of the black-box model prediction to a less critical aspect of the black-box sample weighting.
>
> Answer 5: We have added details of data statistics (including the size of datasets) to Appendix A. Also, in Figure 3 (right), we already report the average selection probability with respect to the lambda in three synthetic datasets, which can be translated into the number of selected samples by multiplying with the number of entire samples for each lambda value considered.

---

### Official Review · AnonReviewer1 · 2019-10-24
**Official Blind Review #1**

**Rating:** 3

**Review:**

In this paper, the authors aim to learn a locally interpretable model via the reinforcement learning approach, to address the fundamental challenge which is that the previous locally interpretable model has smaller representation capacity than black-box models, and causes under-fitting with conventional distillation techniques. Overall speaking, the paper is well organized, and the proposed approach is well tested, but in my opinion, there is a conceptual error.

You claimed your method is REINFORCEMENT LEARNING based, but the REINFORCEMENT LEARNING definition for your task is weird, or wrong. In section 3, you didn't give an explicit explanation for the state transition. With your given RL-like objective function, it seems that the state transition is from features to features. However, there is no specific correlated explanation in your paper on why you make such an assumption. Besides, the state transition in RL relies on decision making at each time step, while it has not reflected in your paper and code, namely, the state-transition independents on the decision making.

To sum up, I don’t think the proposed method is RL-based, it would be more appropriate to define it as a MAB problem, and this paper should solve this problem before publishing.


**Experience Assessment:**

I do not know much about this area.

**Review Assessment: Checking Correctness Of Derivations And Theory:**

I carefully checked the derivations and theory.

**Review Assessment: Checking Correctness Of Experiments:**

I carefully checked the experiments.

**Review Assessment: Thoroughness In Paper Reading:**

I read the paper at least twice and used my best judgement in assessing the paper.

---

> ### Author Response · Authors · 2019-11-13
> **RE: Official Blind Review #1**
>
> Thanks for your insightful comments on our paper. We clarify the question about the reinforcement learning definition we use and its connections to contextual bandits below – we hope that is helpful.
>
> The definition for Reinforcement Learning (RL) frameworks that we use in this paper is from [1] and states that RL requires three components: state, action, and reward, and given the state, the agent selects the action and the environment returns the reward based on both state and action.  The state transition is not a necessary component of a general RL framework because multi-armed bandit, contextual bandit, and one-step RL are all types of RL problems without the state transition.  This is consistent with [1].
>
> In RL-LIM, the three necessary components of an RL framework exist: The state is the vector of input features, the action is the selection vector, and the reward is the fidelity which depends on the input features and the selection vector. The instance-wise weight estimator model is the agent that outputs the actions based on the state (input features). The environment is comprised of the input feature generating process, as well as the black-box model for the target task. We assume that the input features are sampled from the underlying data distribution independently at each iteration. Given this independence assumption, there is no explicit state transition modelling in the proposed framework.
>
> We agree that this framework resembles the contextual bandit framework (which is a type of a general RL framework) where states are not determined by the previous states or actions [2]. However, the typical objective of the conventional contextual bandit framework is to minimize the regret (the cumulative rewards differences between the agent decisions and optimal decisions); on the other hand, the objective of the proposed framework is to maximize the one-step reward (see Equation (1)). Therefore, we claim that our framework is closer to a one-step RL setting. We have included these discussions in the revised manuscript to better position our framework with the notations of a standard RL framework, and explicitly stating the states, actions and the rewards.
>
> We hope that we have fully addressed your questions and concerns. Since we did not notice you pointing out any other weak points of the paper, we sincerely hope that you would consider raising the score after our clarification. Please let us know if you have any further comments.
>
> [1] Kaelbling, Leslie Pack, Michael L. Littman, and Andrew W. Moore. "Reinforcement learning: A survey." Journal of artificial intelligence research 4 (1996): 237-285.
> [2] https://medium.com/emergent-future/simple-reinforcement-learning-with-tensorflow-part-1-5-contextual-bandits-bff01d1aad9c

---

### Decision · Program_Chairs · 2019-12-19

**Decision:**

Reject

**Comment:**

The paper aims to find locally interpretable models, such that the local models are fit (w.r.t. the ground truth) and faithful (w.r.t. the global underlying black box model).
The contribution of the paper is that the local model is trained from a subset of points, selected via an optimized importance weight function. The difference compared to Ren et al. (cited) is that the IW function is non-differentiable and optimized using Reinforcement Learning.

A first concern (Rev#1, Rev#2) regards the positioning of the paper w.r.t. RL, as the actual optimization method could be any black-box optimization method: one wants to find the IW that maximizes the faithfulness. The rebuttal makes a good job in explaining the impact of using a non-differentiable IW function.

A second concern (Rev#2) regards the interpretability of the IW underlying the local interpretable model.

There is no doubt that the paper was considerably improved during the rebuttal period. However, the improvements raise additional questions (e.g. about selecting the IW depending on the distance to the probes). I encourage the authors to continue on this promising line of search.